# Mathematical Models of Death Signaling Networks

**DOI:** 10.3390/e24101402

**Published:** 2022-10-01

**Authors:** Madhumita Srinivasan, Robert Clarke, Pavel Kraikivski

**Affiliations:** 1College of Architecture, Arts, and Design, Virginia Polytechnic Institute and State University, Blacksburg, VA 24061, USA; 2The Hormel Institute, University of Minnesota, Austin, MN 55912, USA; 3Academy of Integrated Science, Division of Systems Biology, Virginia Polytechnic Institute and State University, Blacksburg, VA 24061, USA

**Keywords:** cell death, apoptosis, necroptosis, ferroptosis, pyroptosis, immunogenic cell death, regulatory networks, death execution pathways, mathematical models, computational analysis

## Abstract

This review provides an overview of the progress made by computational and systems biologists in characterizing different cell death regulatory mechanisms that constitute the cell death network. We define the cell death network as a comprehensive decision-making mechanism that controls multiple death execution molecular circuits. This network involves multiple feedback and feed-forward loops and crosstalk among different cell death-regulating pathways. While substantial progress has been made in characterizing individual cell death execution pathways, the cell death decision network is poorly defined and understood. Certainly, understanding the dynamic behavior of such complex regulatory mechanisms can be only achieved by applying mathematical modeling and system-oriented approaches. Here, we provide an overview of mathematical models that have been developed to characterize different cell death mechanisms and intend to identify future research directions in this field.

## 1. Introduction

Mathematical modeling is a powerful tool that allows one to connect molecular biology to cell physiology by associating the qualitative and quantitative features of dynamical molecular networks with signal–response curves measured by cell biologists [1]. Mathematical and systems-oriented approaches have been successfully applied to describe the dynamics of complex molecular networks that control cell cycle [2,3], nutrient signaling [4], checkpoints [5], signaling dysregulation in cancer [6], and cell death [7,8,9,10,11,12,13]. Systems-oriented mathematical approaches are especially useful for analyzing complex systems that cannot be understood by intuitive reasoning. Undoubtedly, cell death regulation is one such molecular mechanism that cannot be fully understood without mathematical modeling. Here, we provide an overview of mathematical models that have been successfully applied to quantitatively characterize death signaling networks.

Cell death mechanisms are directly involved in regulations of tissue homeostasis, inflammation, immunity, development and other physiological processes [14]. Characterization of new genes and molecular components, involved in signaling pathways by regulating cell death, continues to progress. A detailed characterization of cell death regulation can help identify novel targets and develop effective therapeutic protocols to strike acquired drug resistance in cancer cells. Accurate predictive mechanistic models of complex molecular networks regulating cell death can be used to test the effects of new drugs on the system, and to search for synergistic drug combinations and effective treatment protocols. Different modeling approaches have been already successfully applied to model extensive cell death molecular networks. Ordinary differential equations (ODEs), Boolean logic, pharmacokinetic-pharmacodynamic (PK-PD), Petri nets, agent-based modeling (ABM), cellular automata and hybrid approaches are the common choices available to model molecular mechanisms involved in cell death control, decisions and execution [6,12,13,15,16,17,18,19]. 

Cell death execution is an all-or-none, irreversible process [20]. Mathematically the activation of irreversible cell death can be described by an irreversible bistable switch with a stable survival steady state, a stable death steady state, and a third unstable steady state separating the survival and death states [21,22,23,24]. A pro-death signal can induce cell death by driving the bistable system from the survival to the death state. The transition occurs when the pro-death signal reaches a threshold that corresponds to the limit point bifurcation. Transition in the reverse direction, from death to survival, is impossible because the second limit point bifurcation, where the death steady state vanishes, occurs in the biologically irrelevant negative signal values (i.e., the concentration of a death-inducing ligand or stressor cannot be negative). Therefore, the activation of the cell death execution in such a bistable system cannot be reversed, even if the initial cell death trigger is removed. This mathematical description of the cell death activation is consistent with a threshold mechanism for cell death induction [25] and an all-or-none death decision [22,26,27]. Importantly, understanding how cells control the cell death/survival switch can help to identify targets that can force cancer cells to flip the switch to activate the irreversible cell death execution.

Complexity of cell death regulatory networks, a requirement to account for all important regulating molecular details and pathways, availability of merely small sets of sparse data for model calibration, as well as under- and over-fitting of the model are issues that must be routinely solved in order to develop a predictive model of cell death [6,15]. This review describes mathematical models that have been successfully applied to quantitatively characterize such cell death control mechanisms as apoptosis, necroptosis, ferroptosis, pyroptosis and immunogenic cell death.

## 2. Apoptosis

Apoptosis is one of the most well-studied and characterized programmed cell death mechanisms. The detailed characterization of molecular interactions involved in apoptosis, and the growing amount of related quantitative data, has encouraged computational and systems biologists to develop mathematical models of apoptosis [12,13,17]. Over the last twenty years, several dozen mathematical models of apoptosis regulation have been described. These apoptosis models aim to explain different data or effects of different treatments on cell death. While the core molecular components regulating apoptosis are shared by all models, variations in molecular circuit designs, components, data, mathematical approaches, and study goals make each model a unique tool to study apoptosis. Most often, molecular mechanisms of apoptosis are mathematically represented using ODEs [7,21,22,25,26,27,28,29,30,31,32,33], Boolean logics [34,35,36], and Petri nets [16]; other computational approaches have also been applied [18,37,38].

The execution core of apoptosis regulation involves a family of proteases termed caspases. Caspases can be separated into the following two groups: effector or executioner caspases (caspase-3, -6, -7), and active initiator caspases (e.g., caspases-8, -9). Activation of the caspases initiates the cleavage of several important cellular proteins, such as actin and nuclear lamins, which results in cell body and nuclear shrinkage and cell death [39]. Apoptosis can be processed through mitochondria-dependent (intrinsic apoptosis) and mitochondria-independent (extrinsic apoptosis) caspase-3 activation pathways [14]. The core components involved in these two pathways are commonly included in all mathematical models of apoptosis and can be found in the earliest mathematical models of apoptosis [7].

Extrinsic apoptosis is characterized by high amounts of active caspase-8 that activates the downstream effectors caspase-3, caspase-6, and caspase-7. The activation of caspase-8 is receptor-mediated, which occurs upon receipt of a death signal that is processed by a surface death receptor such as FAS (a member of the tumor necrosis factor gene superfamily) [14]. Therefore, extrinsic apoptosis is a receptor-mediated cell death mechanism, as shown in Figure 1 (left). By contrast, intrinsic apoptosis can be executed even in cells with lower levels of active caspase-8 but requires an additional amplification that involves activation of the pro-apoptotic functions of the mitochondria. For example, stress-related factors (e.g., DNA damage) can induce activation of the executioner caspases via a mitochondria-dependent pathway in the absence of an external death signal [40] (Figure 1, right panel). The mitochondria-dependent pathway begins with the cleavage of anti-apoptotic Bcl-2 family members, which causes the aggregation of pro-apoptotic proteins such as Bax and Bak. Aggregation of pro-apoptotic proteins is followed by the release of cytochrome *c* from the mitochondria, which induces the formation of a large protein complex known as the apoptosome. The apoptosome recruits and activates caspase-9, allowing it to cleave the downstream effectors pro-caspase-3, pro-caspase-6, and pro-caspase-7. Notably, the expression of anti-apoptotic Bcl-2 family members can block the intrinsic apoptosis signaling in cells. By contrast, extrinsic apoptosis cannot be blocked by the expression of high levels of Bcl-2 proteins because large amounts of caspase-8 are already generated. 

The earliest mathematical models of apoptosis described both mitochondria-dependent and independent death activation pathways. In early 2000, Fussenegger et al. published a mechanistic ODE-based mathematical model of apoptosis that describes both receptor-mediated and stress-induced caspase activation mechanisms [7]. The receptor-mediated feature of the model describes the FAS surface receptor that activates procaspase-8. Activation of apoptosis initiator caspases involves the following reactions: the binding of an extracellular death ligand to the FAS receptor, the binding of FAS-associated death domain (FADD) protein to the FAS death domain, and the binding of caspase-8 to a domain on FADD that enables caspase-8 activation by proteolytic cleavage. Each binding process is described by a specific rate parameter in the model. Simulation results show that about 50% of procaspase-8 is activated within two hours after the death signal is received. After procaspase-8 activation, the executioner caspase is activated within minutes, and then the initiation of procaspase-9 occurs with the lag time ~20–30 min. The activation curves have a sigmoidal shape indicating, that the transition between the inactive to the active state is characterized by a threshold. If the binding between FADD and clustered FAS death domains is disrupted, then only <0.1% of active caspase-8 is observed upon receipt of the death signal, which is consistent with experimental observations [41].

Fussenegger’s model of stress-mediated apoptosis regulation describes the activation of procaspase-9 by cytosolic cytochrome *c*, and the apoptotic protease-activating factor 1 (Apaf-1) complex. Activated caspase-9 then activates apoptosis executioner caspases at some specific rate. Formation of the Apaf-1–cytochrome *c* complex is inhibited by antiapoptotic Bcl-2 family members such as Bcl-x_L_. Proapoptotic Bcl-2 family members (e.g., Bax, Bak) can bind to antiapoptotic family members and remove their inhibitory effect. The ratio of anti- versus pro-apoptotic Bcl-2 family members is controlled by the p53 transcription factor that is activated in cells under stress conditions. Simulation results of stress-induced caspase activation dynamics were consistent with experimental observations [42]. Specifically, the model shows that cytochrome *c* is released within 10 min after a stress death signal is received, which results in procaspase-9 activation, 35–40% of the executioner caspase being active within 1 h, and 70% of the executioner caspase being active at 2 h. In addition, simulations revealed that the active fraction of both initiator and executioner caspases is reduced in p53 mutant cells as compared to wild-type cells. Overexpression of antiapoptotic Bcl-2 family members is predicted to block the activation of procaspase-9. The model also confirms that the ratio of anti- versus pro-apoptotic Bcl-2 family members determines whether or not executioner caspases will be activated. The model was then used to predict the effects of combined therapies based on simultaneous receptor- and stress-induced caspase activation.

The model developed by Fussenegger et al. was successful in explaining qualitative experimental observations. However, more quantitative data would be required to complete the model calibration. Quantitative information on reaction rates and molecular concentrations is required to perform reliable mathematical simulations of signal transduction in the apoptosis regulatory network. In 2004, Eissing et al. developed a reduced receptor-induced apoptosis, using parameter values from the literature to evaluate the system behavior within a wide range of parameters [21]. The model revealed that caspase activity remains low for a time that is inversely proportional to the stimulus strength, followed by a steep rise in activity when the input exceeds the threshold; caspase activity then ceases at some maximum level. Bifurcation analysis of the model confirmed that the apoptosis regulation system exhibits a bistable behavior. The same year, Bentele et al. developed a data-based model of receptor-induced apoptosis with parameters estimated on the basis of quantitative experimental data [25]. The time series data for concentrations of 15 different molecules after activation of FAS receptors were used to calibrate the core model of the FAS-induced apoptosis. In addition, data from distinct apoptosis activation scenarios in response to different initial values of ligand concentration were used to improve the estimation of model parameters. The model predicted that apoptosis is not executed when a ligand–receptor concentration ratio is below a critical value, which was also confirmed by experimental observations. In conclusion, Bentele et al. proposed a threshold mechanism for induction of receptor-induced apoptosis. A year later, Hua et al. published a FAS-induced apoptosis model to investigate the effects of altering the level of Bcl-2 on the kinetics of caspase-3 activation [43]. The model predicts that Bcl-2 blocks the mitochondrial pathway by binding to proapoptotic Bax, Bak, and tBid proteins. Further, the model predicts that apoptosis signaling flow can be switched between mitochondria-dependent and mitochondria-independent pathways by varying molecular component levels without changing network structure.

In 2006, Legewie et al. developed a quantitative kinetic model of intrinsic (stress-induced) apoptosis, which displays an all-or-none behavior of caspase activation in response to an apoptotic stimulus [22]. The model helped to identify the positive feedback mechanism that allows cells to achieve ultrasensitivity and bistability in cell death decision making. The pathway molecular regulators that control the apoptotic threshold stimulus and integrate multiple inputs into an all-or-none caspase output were also determined. Time-course simulation results agreed with experimental observations that the induction of maximal caspase-3 cleavage after exogenous addition of cytochrome *c* occurs within ~15–60 min. Furthermore, cytochrome *c*-induced activation of caspase-3 was observed to be bistable and irreversible. The bistable and irreversible caspase-3 activation arises in the system due to XIAP-mediated feedback that cooperates with caspase-9 cleavage by caspase-3. X-linked inhibitor of apoptosis (XIAP) inhibits the catalytic activities of caspase-9 and caspase-3 through reversible binding. The feedback cleavage of caspase-9 by caspase-3 leads to autoamplification of the apoptotic signal. Simulation results show that XIAP-mediated feedback is observed only if caspase-9 and caspase-3 compete for binding to XIAP. Depletion and re-addition experiments using different Apaf-1, caspase-3, caspase-9, and/or XIAP concentrations were proposed to test the all-or-none caspase activation. 

Also in 2006, Rehm et al. published a computational model of apoptosome-dependent caspase activation based on biochemical data from HeLa cells [26]. The model predicts that the all-or-none apoptotic response depends on caspase-3-dependent feedback signaling and XIAP, which was then verified quantitatively using single-cell experiments with a caspase fluorescence resonance energy transfer substrate. A concentration threshold of XIAP between 0.15 and 0.30 μM, controlling the substrate cleavage by effector caspases, was identified. The model suggested that high levels of XIAP may promote apoptosis resistance and sublethal caspase activation. This result agrees with a computational analysis that was performed earlier, which also suggested that the inhibitor of apoptosis plays an important role in both the induction and prevention of apoptosis [44]. Conversely, Bagci et al. proposed a mathematical model of mitochondria-dependent apoptosis to study both the role of Bax and Bcl-2 synthesis, degradation rates and the number of mitochondrial permeability transition pores involved in the cell response to a death signal [23]. The main finding was that the transition from bistable to monostable (survival) cell behavior is controlled by the synthesis and degradation rates of Bax and Bcl-2 and by the number of mitochondrial permeability transition pores. Also, the model results suggested that cooperative apoptosome formation is a much more robust mechanism to induce bistability than feedback mechanisms involving, for example, the inhibition of caspase-3 by the inhibitor of apoptosis. Later, Chen and Cui et al. analyzed the robustness of Bax and Bcl-2 apoptotic switches using both deterministic and stochastic models [38,45,46]. These mechanisms were confirmed to be bistable and robust to noise and wide ranges of parameter variation.

Albeck et al. developed a mathematical model of extrinsic, receptor-induced apoptosis to explain the molecular mechanism of the variable-delay, snap-action switch function that determines the cell choice between life and death [27]. The model was calibrated by experimental data collected from live-cell imaging, flow cytometry, and immunoblotting of cells perturbed by protein depletion and overexpression. The model was then used to reveal the mechanism by which a steady and gradual increase in caspase-8 activity is converted into a snap-action downstream signal. Permeabilization of the mitochondrial membrane and relocalization of proteins are the key factors in the extrinsic apoptosis network by which a graded signal that activates caspase-8 and promotes the formation of pores in the mitochondrial membrane is transformed into an all-or-none death decision. Importantly, such snap-action behavior at the level of the mitochondrial outer membrane permeabilization occurs independently of caspase-dependent feedback mechanisms. The formation of pores in the mitochondrial membrane involves the pore-forming proteins Bax and Bak that can self-assemble into transmembrane pores, which are antagonized by anti-apoptotic Bcl-2 proteins [47]. Cytochrome *c* is released into the cytosol when the level of active pore-forming proteins exceeds the threshold set by anti-apoptotic Bcl-2 proteins. Using experimental and modeling techniques, Spencer et al. demonstrated that cell-to-cell variability in time-to-death significantly depends on the activation rate of the tBid protein that activates the pore-forming proteins, Bax and Bak [33]. Therefore, in the case of receptor-mediated apoptosis, the timing and probability of death relies on the differences in the protein levels that can be caused, for example, by noise in gene expression. Furthermore, the stochastic protein turnover in a receptor-mediated apoptosis model can result in fractional killing [48].

Later models were developed to investigate crosstalk between apoptosis regulation and NF-κB pathways [32], the estrogen signaling network [31], endoplasmic pathways [28], and autophagy regulation [29]. Neumann et al. described a model of the crosstalk between receptor-mediated apoptosis regulation and NF-κB signaling that are activated by the same receptor in parallel to the apoptotic signaling and on a similar time scale [32]. Model and experimental analysis suggested that the balance between apoptotic and NF-κB signaling is shaped by the proteins that regulate the assembly dynamics of the death-inducing signaling complex (DISC). Therefore, the assembly of DISC acts as a signal processor, determining life/death decisions in a nonlinear manner. Tyson et al. provided a roadmap for a detailed mathematical model that would allow researchers to characterize the crosstalk among the estrogen signaling network, apoptosis, autophagy, and cell cycle regulations in breast epithelial cells [31]. Later, the same research lab published a detailed mathematical model to examine the decision process that moves a cell from autophagy to apoptosis [29]. The model was successful in explaining quantitative time-course data of autophagy and apoptosis under cisplatin treatment. Further, the model allows for characterization of the prosurvival and prodeath cell responses to cytotoxic stress. Also, in 2012, Hong et al. published a model of cisplatin-induced apoptosis that integrates the death receptor pathway, and mitochondrial and endoplasmic reticulum stress response mechanisms [28]. The model predicts the relative contribution of each signaling pathway to apoptosis. Simulation results revealed that the mitochondrial and death receptor pathways as well as crosstalk among pathways make the greatest contribution to the level of apoptosis, whereas the contribution of the endoplasmic reticulum stress pathway is negligible.

### The Role of p53 in Apoptosis

The tumor suppressor gene p53 (TP53) has been reported as an upregulated modulator of apoptosis and as a driver of cell fate transition from cell cycle arrest to apoptosis [49]. Mathematical models that characterize the p53 contribution to apoptosis have been developed by several groups [7,23,28,30,50]. p53 targets many genes regulating cell apoptosis, including BCL2 and BAX genes [51]. Computational study of apoptosis regulation shows that the balance between anti- and proapoptotic Bcl-2 family members is altered in p53 mutant cells [7]. Also, the active fraction of both initiator and executioner caspases is reduced in p53 mutant cells as compared with wild-type cells. The mathematical model also predicts that overexpression of the death ligand and the FAS receptor can be used to initiate executioner caspase activation in p53 mutant cells [7]. Bagci et al. have shown that apoptosis is not sensitive to caspase-3 activation when p53 expression is low, and that bistability to apoptotic stimuli is observed when p53 level is high [23]. Predictions from this apoptosis model agree with experimental data [52]. Another study reported that inhibition of p53 protects against cisplatin-induced apoptosis [28]. Cisplatin induces DNA damage that results in the phosphorylation and activation of p53. There, the activation of Bax by p53 induces mitochondrial membrane permeabilization and apoptosis [53]. Also, p53 mediates caspase-2 activation and the mitochondrial release of apoptosis-inducing factor. The model predicts time courses for p53, caspase-2, Bax activation, apoptosis-inducing factor release and apoptosis activation. Simulation results agree with experimental data that p53 inhibition prevents the mitochondrial release of apoptosis-inducing factor and cisplatin-induced apoptosis [54]. Overexpression of p53 results in caspase-2 activation and also the mitochondrial release of apoptosis-inducing factors [54].

Ballweg et al. developed a mathematical model that integrates p53 signaling, cisplatin-induced events, and apoptosis regulation that was used to study the dynamics of fractional killing induced by cytotoxic drugs [30]. Many drugs activate not only apoptosis execution signaling but also expression of anti-apoptotic genes, which results only in fractional killing amongst a population of treated cells [55]. Thus, fractional killing may occur due to crosstalk between the apoptosis and survival pathways [56]. The model predicts that the probability of apoptosis depends on the dynamics of p53 and the rate of p53 activation determines the cell fate [30]. Slow activation of p53 results in cell survival, whereas fast p53 activation induces cell death. This result also agrees with the experimental observation showing that apoptotic cells accumulate p53 much earlier than cells that survive the treatment [55]. In the model, activation of Bax and subsequent execution of apoptosis occur when the level of p53 exceeds a threshold value. However, the apoptosis initiation threshold depends on the inhibitor of apoptosis, cIAP. Cells with an elevated level of cIAP require a higher level of p53 to induce apoptosis. Because the level of apoptosis regulator cIAP increases with time, the rate of p53 activation plays an important role in the determination of cell fate. Cell-to-cell variability due to stochastic gene expression and environmental noise can also set different apoptosis initiation thresholds in different cells, resulting in fractional killing.

Up to this point, we have reviewed mathematical models of apoptosis that use ODEs to describe the mechanism of cell death (apoptosis) regulation. However, other mathematical approaches have been also used to study apoptosis regulation [16,18,34,35,36,37]. Several apoptosis models have been developed using a Boolean (logical) approach that can analyze extensive regulatory networks with many molecular components and their interactions [34,35,36]. Schlatter et al. developed an apoptosis regulation model that comprises 86 nodes and 125 interactions [34]. Mai et al. developed a model that describes 37 internal states of signaling molecules involved in apoptosis regulation, 2 extracellular signal inputs, and the DNA damage event as an output [35]. Calzone et al. developed a model to study crosstalk between receptor-mediated apoptosis regulation, NFκB pro-survival pathways, and RIP1-dependent necroptosis regulation [36]. These models were used to characterize feedback loops in the apoptosis regulation network structure.

While Boolean models are excellent tools to reproduce the *qualitative* behavior of a regulatory network, they are weak at addressing detailed *quantitative* questions about molecular mechanisms [19]. Petri nets have been applied to analyze and validate a qualitative model of extensive apoptosis regulation [16]. Agent-based modeling turned out to be a more appropriate approach for modeling the death-inducing signaling complex assembly than an ODE-based model that must describe a large number of intermediate products involved in DISC assembly [37]. A cellular automata approach has been applied to study apoptosis blocking in the immunological response of T cells by varying the inhibitor actions such as FLIP and IAP [18]. The model predicts that only joint suppression of both FLIP and IAP apoptosis inhibitors can effectively act to kill cancer cells through apoptosis.

In conclusion, comprehensive data and extensive experimental characterization of apoptosis allowed computational and systems biologists to develop several mathematical models of apoptosis regulation. These models not only increase our understanding of mechanisms of apoptosis execution induced by stress or signals, but also predict perturbations that can prevent or enhance apoptosis. An accurate mathematical model of apoptosis can help find novel combinations of existing therapies that can induce the death of cancer cells using low doses. Further studies that integrate apoptosis with other cell death regulations will help to understand the cell death decision mechanism that determines the execution of a specific cell death fate. 

## 3. Necroptosis

Necroptosis is a regulated cell death that can be initiated by changes in extracellular or intracellular homeostasis, detected by specific death receptors [14]. Triggering necroptosis primarily involves the receptor-interacting protein kinase 1 (RIP1), RIP3, and mixed lineage kinase domain-like protein (MLKL). Necroptosis can be induced by the binding of tumor necrosis factor (TNF) or other ligands to cell surface receptors that trigger the sequential phosphorylation of receptor-interacting protein kinases. At a cell physiology level, necroptosis results in cell volume expansion, cell membrane rupture, and intracellular material overflow that leads to a local inflammatory reaction and immune response activation. Necroptosis-inhibiting drugs can be used to treat inflammatory diseases [57]. Necroptosis-promoting drugs are potential anticancer therapies [58]. Studies of necroptosis regulation can help to identify molecular targets that can be used to reprogram the necroptosis execution in a desired direction. While many molecular components involved in necroptosis regulation are known, the precise interaction network, signaling spread, dynamic behavior of necroptotic regulation, and the decision-making processes within the molecular network, remain poorly understood. Several mathematical models have been developed recently to characterize the dynamics of necroptosis regulation [8,59].

Xu et al. have developed a computational model of the cellular necroptosis signaling network [8], to study the necroptosis signaling dynamics that lead to cell death in the form of oscillation-induced trigger waves. The study focused on the core cellular necroptosis signaling module that includes four components: TRADD, RIP1, caspase-8, and RIP3. The activities of key components are regulated either by phosphorylation, dephosphorylation, or cleavage. The corresponding mathematical model described 4 variables and involved 10 interaction terms. Xu et al. used a Latin hypercube sampling method to randomly scan the model network parameters and evaluate the stable oscillation behavior of the cellular necroptosis signaling circuit. Bifurcation analysis and potential landscape theory were applied to explore oscillation modes in different cellular necroptosis signaling circuits. The results indicate that the cellular necroptosis signaling circuit more likely produces oscillations when the total amount of RIP1 or caspase-8 is high, while fluctuations in the value of RIP3 have no significant effects on the oscillation probability. Also, oscillations are often obtained when the activation of caspase-8 by RIP1 is fast, while RIP3 phosphorylation by RIP1 is relatively slow. Further, oscillations are more robust when the reaction rate constants that describe RIP1 activation by RIP3 are stronger than rate constants describing other interactions. Overall, oscillation robustness analysis revealed three regulatory feedback loops formed by RIP1, caspase-8, and RIP3 interactions. These loops comprise a negative feedback loop: RIP3 activates RIP1, which activates caspase-8, that inhibits RIP3; a positive feedback loop: RIP1 activates RIP3, which inhibits caspase-8, that inhibits RIP1; and an incoherent feedforward loop: RIP1 activates both caspase-8 and RIP3, and caspase-8 inhibits RIP3. Importantly, for oscillations to be robust, the reactions in the positive feedback loop must be slower than reaction rates in the negative feedback loop. Also, a stochastic parameter analysis indicated that the incoherent feedforward loop is the essential molecular mechanism that allows the necroptosis signaling system to generate oscillations.

Xu et al. classified oscillations that occur in cellular necroptosis signaling circuits into four groups according to amplitude and oscillation period. About 50% of observed oscillations had a high-amplitude (above the median value of all the counted amplitudes) and fast period (>100 min based on the oscillation period of NF-κB [60,61]), about 37% of oscillations had a low-amplitude and fast period, ~12% of oscillations had high-amplitude and slow period, and ~1% of oscillations had a slow and low-amplitude period. Further analysis revealed that the inhibition rates of RIP1 and RIP3 by caspase-8 play an important role in determining the amplitude behavior of fast oscillations. In addition, bifurcation analysis revealed that the dynamic behavior of the system can be switched from slow high-amplitude oscillations to slow low-amplitude oscillations by tuning the parameters that describe the activation of caspase-8 by RIP1. However, the transition from fast to slow oscillation behavior cannot be achieved by changing any single reaction rate constant. Also, the system changes dynamics from slow high-amplitude oscillations to fast low- or high-amplitude oscillations when two parameters that describe RIP1 inhibition with caspase-8 and RIP1 phosphorylation with RIP3 are simultaneously tuned. Robustness analysis revealed that the period of fast oscillations was more robust to parameter perturbations than the period of slow oscillations. The amplitude of slow low-amplitude oscillations was robust to parameter perturbations, while the robustness of amplitude of fast high-amplitude oscillations was the weakest. Overall, the study provides a quantitative characterization for the mechanism of oscillation mode-switching behavior in the necroptosis signaling network. Xu et al. proposed that MLKL can decode the information according to the amplitude and period of RIP3, which can be an important mechanism that allows cells to generate different responses in various stressful conditions.

A more recent detailed computational model of tumor necrosis factor (TNF)-induced necroptosis has been developed by Ildefonso et al. [59]. The model was derived from the literature-curated molecular mechanisms of necroptosis regulation, which involves 14 proteins, 37 biochemical species, and 40 reactions. The simplified molecular mechanism that shows key species involved in necroptosis execution is shown in Figure 2. Dynamics of species were described by a set of ordinary differential equations where all reactions were described by the mass action law. The model was calibrated and validated using experimental protein time-course data from a well-established necroptosis-executing cell line. Simulations then confirmed that the model is successful in explaining the dynamics of necroptosis reporter, phosphorylated mixed lineage kinase domain-like protein (pMLKL). Furthermore, four distinct necroptosis execution modes were identified by using a dynamical systems analysis and a spectral clustering algorithm. While the temporal dynamics of pMLKL were similar in each mode of necroptosis execution, the sequences of molecular events that led to MLKL phosphorylation and subsequent necroptotic cell death were different. The modes primarily differed in the values of rate constants across the necroptosis execution pathway. For example, the rate constant for binding of A20 to ubiquitinated RIP1 was significantly smaller in mode 4 than in the other modes, and also smaller in mode 2 relative to modes 1 and 3. Also, mode 4 has a significantly larger activation rate and smaller deactivation rate constant for caspase-8 in complex II. The activation/deactivation of caspase-8 in complex II is a critical step in the pathway for determining whether the cell will progress to necroptosis. Differences in rate constant values create the difference in the action of A20 and CYLD enzymes across four modes that are then able to effectively operate as an inhibitor or activator of necroptosis. 

Taken together, the computational analysis helped to resolve the controversy in experimental observations by showing that CYLD- and A20-driven deubiquitination of RIP1 may act as pro- and anti-necroptotic in different cell types. According to Ildefonso et al.’s model, knocking out A20 decreases the probability of necroptosis execution (necroptosis sensitivity) in mode 1, and increases the sensitivity to necroptosis in mode 2 [59]. Conversely, knocking out CYLD increases the sensitivity to necroptosis in modes 1 and 4, and decreases the sensitivity to necroptosis in mode 2. Knocking out CYLD or A20 has no effect in mode 3. Also, A20 knockout has no effect in mode 4. These results have been compared to cell phenotype observations in A20 and CYLD knockdown experiments in different cell types. For example, it has been reported that RIP1 is deubiquitinated by both A20 and CYLD in mouse fibrosarcoma cells, but inhibition of CYLD protects cells from necroptosis, while A20 depletion can sensitize cells to death by necroptosis [62]. Thus, A20 and CYLD depletion experiments in mouse fibrosarcoma cells are consistent with the model results obtained for A20 and CYLD knockouts in mode 2.

TNF, TNFR, and MLKL are three common protein modulators of necroptosis across the four modes of necroptosis execution. Furthermore, rate constants that control the association of TNF to TNFR, ubiquitination of RIP1 by cIAP in complex I, and association of LUBAC to complex I can be used to efficiently modulate necroptosis execution across the four modes. Therefore, targeting these modulators can be used to induce or prevent necroptosis, potentially useful for both cancer therapy and treatment of inflammatory diseases.

Apoptosis and necroptosis regulation networks share common nodes and edges and may suppress each other [63]. Either apoptosis or necroptosis can be induced by TNF and the cell death decision depends on the cell state. Complex II can recruit RIP3 to form a necrosome or recruit caspase-8 to stabilize its active conformation, resulting in the release of an activated caspase 8 homodimer that then can induce apoptosis [64]. Li et al. [65] performed a quantitative study of crosstalk between the apoptosis and necroptosis pathways. Specifically, mathematical modeling was used to investigate three possible mechanisms of caspase-8 activation by (i) TRADD, (ii) RIP1, and (iii) TRADD and RIP1 together. The law of mass action was used to convert the proposed molecular mechanisms into a system of ODEs. Simulations of each mechanism were compared with data obtained using the sequential window acquisition of all theoretical fragment ion spectra mass spectrometry methods. All three mechanisms reproduced the amounts of major components in TNFR1, RIP1, and RIP3 complexes. However, only mechanism (ii) could explain a negative regulation of RIP3 phosphorylation by the increase in RIP1 levels. This result was also supported by a sensitivity analysis showing that the most robust negative regulation of RIP3 phosphorylation by RIP1 is achieved when mechanism (ii) is used in the model. To test this prediction, Li et al. experimentally knocked down RIP1 to three different expression levels by using RIP1-specific short hairpin RNA and measured the increase in TNF-induced phosphorylation of RIP3 and MLKL. Deletion of RIP1 completely blocks TNF-induced RIP3 phosphorylation [65]. In addition, simulation results show that pro-caspase-8 activity is necessary for the up-regulation of RIP3 phosphorylation by decreasing RIP1 expression. The mechanism was further refined to make it in agreement with the observation that TNF induces quick caspase-8 activation and apoptosis in RIP1 KO cells [62]. Specifically, TRADD-dependent caspase-8 activation was added to the mechanism (ii). The final model successfully explained both RIP1′s biphasic roles in necroptosis, where RIP1 promotes necroptosis within an extremely low-level range (<∼2% of wildtype) and inhibits necroptosis at higher levels, and the activation level of caspase-8 in RIP1 KO cells. Also, the response of pro-caspase-8 to RIP1 level is linear, whereas RIP3 phosphorylation is determined by the nonlinear (ultrasensitive) threshold pattern.

Overall, a quantitative approach has been applied successfully to describe the roles of RIP1 in cell death determination. In conclusion, Li et al. proposed a “speed competition” decision mechanism in which cells decide to execute apoptosis or necroptosis by the pathway that reaches the final destination first. Interestingly, simultaneous execution of necroptosis and apoptosis has been observed in some individual cells [65]. 

## 4. Pyroptosis

The regulated cell death that is associated with the formation of plasma membrane pores by members of the gasdermin protein family is called pyroptosis [14]. The induction of pyroptosis may occur as a consequence of inflammatory caspase activation that can be triggered by pathogen invasion such as Gram-negative bacteria. The critical role of caspase-driven pyroptosis for innate immune responses against invading bacteria has been confirmed in experiments with mice carrying gene mutations that disrupt normal activity of caspase proteins [66]. By killing the host cell, pyroptosis removes the replication compartment of intracellular pathogens and thus prevents their spreading. Hence, pyroptosis has an important role in innate immunity against intracellular pathogens. 

Pyroptosis induced by inflammatory caspases is driven by the gasdermin protein GSDMD. Caspases activate GSDMD that then translocates to the plasma membrane where GSDMD induces pore formation and thus rapid plasma membrane permeabilization. The simplified molecular mechanism of the pyroptosis induced by inflammatory caspases is shown in Figure 3. In this scheme, pyroptosis relies on caspase-1 activation. 

Beyond inflammatory settings, pyroptotic cell death can be induced by TNF, various DNA-damaging agents, and infection with vesicular stomatitis virus [67,68]. In these cases, pyroptosis is driven by other members of the gasdermin family, specifically GSDME. This form of pyroptosis releases fewer inflammatory cytokines than is observed when pyroptosis is induced by inflammatory caspases. Pyroptotic signaling relies on the activation of caspase-3 that catalyzes proteolytic cleavage of GSDME. The identification of other gasdermin family members that execute pyroptosis in conditions that are beyond inflammatory settings has been significantly expanded [14].

A computational study of the crosstalk between caspase-1- and caspase-3-driven pyroptosis pathways was performed by Zhu et al. [9]. The molecular regulatory network that executes pyroptosis via activation of GSDMD and GSDME is shown in Figure 3. The crosstalk between caspase-1- and caspase-3-driven pyroptosis pathways is realized through tBid, caspase-9, and caspase-8 components. Zhu et al. developed a mathematical model that describes the dynamics of seven molecular components and the dynamics of the cell population governed by cell proliferation and death processes. The model consists of eight coupled ODEs and 83 parameters. Hill functions were used to describe activation and inactivation reactions for molecular components. The values of 44 parameters were estimated from sources available in the literature and 39 parameters were estimated using 138 time-course data points that were measured for eight variables (the death rate and seven molecular components) in wild-type cells and cells with single, double, and triple knockouts of the molecular components. 

The pyroptosis decision mechanism was analyzed using bifurcation and sensitivity analysis methods. Bifurcation analysis revealed that the change in expression levels of caspase-1, caspase-3, and GSDMD can switch between GSDMD- and GSDME-executed pyroptosis death modes. Furthermore, the transition between pyroptosis death modes could not be efficiently controlled by varying the expression levels of caspase-8, caspase-9, tBid, or GSDME. According to the model, GSDMD-driven pyroptosis is more likely when the caspase-1 total expression level is below ∼1.5 nM and GSDME-driven pyroptosis occurs when the caspase-1 level is above 14 nM. For caspase-1 levels ranging from 1.5–14 nm, bistability is observed when either GSDMD- or GSDME-driven pyroptosis may occur. Similarly, when GSDMD level is lower than 88 nM, GSDME-driven pyroptosis is induced, whereas cells can selectively execute either pyroptosis mode when the level of GSDMD is between 88 nM and 165 nM. GSDMD-driven pyroptosis occurs when GSDMD level is higher than 165 nM. Also, cells execute GSDMD-driven pyroptosis when caspase-3 level is lower than 250 nM, and selectively induce either GSDMD- or GSDME-executed pyroptosis with higher levels of caspase-3.

Sensitivity analysis confirmed that the expression levels of GSDMD and caspase-1 can efficiently change the pyroptosis death modes. This result agrees with experimental observations [69,70]. In addition, bifurcation analysis predicts that the expression level of caspase-3 can also change the pyroptosis death mode between caspase-1- and caspase-3-driven pyroptosis. Overall, the model predicted 3 molecular components and 12 reactions that can be targeted to control the switch between modes of pyroptosis execution. Drugs that can switch between pyroptosis death modes can help to improve treatment protocols for cancer and inflammasome-mediated diseases. For example, GSDME-induced pyroptosis can act as a tumor suppressor [71,72] and also releases fewer inflammatory cytokines when compared to pyroptosis that is executed by GSDMD.

Li et al. extended the GSDMD-induced pyroptosis model by adding apoptosis regulation [73]. The model allows one to study the crosstalk between pyroptosis and apoptosis and inflammasome-induced cell death under different perturbation conditions. Simulation results reproduce the dynamics of cell death executioners in multiple knockout cells. Pyroptosis and apoptosis events are determined by the level of cleaved GSDMD and cleaved caspase-3, respectively. Sensitivity analysis was performed to determine the molecular components that can significantly affect the occurrence of pyroptosis and apoptosis. The model predicted that caspase-1 and GSDMD are key molecular regulators directing the signal flow that can switch cell death modes between pyroptosis and apoptosis. Decreases in caspase-1 or GSDMD gradually inhibit pyroptosis and enhance apoptosis induction. These model predictions were validated by caspase-1 and GSDMD-knocked down experiments. Furthermore, the model results helped to suggest the death signal propagation pathways, resulting in pyroptosis or apoptosis in cells expressing different levels of caspase-1 or GSDMD. To understand the roles of caspase-1 and GSDMD in triggering the cell death modes, Li et al. employed a potential landscape approach. The cell death landscape was represented by potential wells corresponding to pyroptosis and apoptosis death modes. In the double-well potential landscape, the system evolved into one of the two wells from any initial condition. Caspase-1 or GSDMD could change the potential landscape from monostable to bistable. A monostable landscape corresponding to pyroptosis is obtained in cells with a high expression level of caspase-1 or GSDMD; the potential landscape changes to bistable and then to an apoptotic monostable as the expression level of caspase-1 or GSDMD decreases. Overall, the model helps to understand the inflammasome-induced cell death, crosstalk between pyroptosis and apoptosis, and may be used to determine potential molecular targets for driving cells into a desired death execution mode.

## 5. Ferroptosis

Ferroptosis is another regulated cell death mechanism that involves iron-catalyzed lipid damage [14,74,75]. Cell death occurring by ferroptosis correlates with the accumulation of markers of lipid peroxidation and can be suppressed by iron chelators, inhibitors of lipid peroxidation, and lipophilic antioxidants [75]. Ferroptotic cell death can be modulated pharmacologically and genetically by perturbing lipid repair systems that involve glutathione and glutathione peroxidase 4 (GPX4) that convert toxic lipid hydroperoxides (L-OOH) into non-toxic lipid alcohols (L-OH) [76]. Depletion or inactivation of GPX4 results in overwhelming lipid peroxidation that causes cell death. Ferroptosis also depends on a set of enzymatic reactions that regulate the biosynthesis of membrane polyunsaturated fatty acids (PUFA)-containing phospholipids, which are the substrates of pro-ferroptotic lipid peroxidation products [75]. Also, the formation of coenzyme-A-derivatives of PUFAs (PUFA-CoA) and their insertion into phospholipids are necessary for the induction of a ferroptotic death signal. Two enzymes, ACSL4 and LPCAT3 are involved in the biosynthesis and remodeling of PUFAs [75,77]. Depletion of PUFAs can suppress the occurrence of ferroptosis, and loss of ACSL4 and LPCAT3 gene products increases resistance to ferroptosis [75]. 

Iron induces the accumulation of lipid peroxides and thus is important for the execution of ferroptosis. Intracellular iron levels depend on the iron efflux pump ferroportin and the iron importer TFR1 and other proteins that regulate iron import, export, and storage [78,79,80]. Also, for ferroptosis to start, phospholipid molecules containing polyunsaturated fatty acids (LH-P) are formed from PUFA-CoA, which are then oxidized into lipid hydroperoxides (L-OOH) and eventually into lipid radicals (LO*). LH-P generation is regulated by LPCAT3 and inhibited by monounsaturated fatty acids (MUFAs). Production of MUFAs depends on desaturation of the saturated fatty acids (SFAs) which is catalyzed by the desaturase SCD1 [81]. Formation of lipid radicals LO* is promoted by reactive oxygen species (ROSs) and lipid peroxidation enzymes including ALOX15 [74,82]. The generation of endogenous lipid radicals initiates ferroptosis. In addition, ferroptotic cell-death responses can be modulated by p53 activity [83]. For example, induction of SAT1, a transcription target of p53, is correlated with the expression levels of ALOX15 [83]. The influence diagram that reflects the molecular mechanism of ferroptosis is shown in Figure 4. Overall, ferroptosis is morphologically and mechanistically different from all other types of regulated cell death. Regulation of ferroptosis has great potential for cancer therapy, and molecular targets that promote ferroptosis are being actively explored [84].

Konstorum et al. developed a stochastic, multistate, discrete mathematical model of ferroptosis regulation [10]. The model describes states of eleven variables that represent ALOX15, GPX4, L-HP, LIP, LO*, L-OOH, LPCAT3, MUFA, PUFA-CoA, ROS, and SLC7A11. Each variable can take on three values that respectively represent low, medium, and high molecular species activity or expression level. Variables are updated using updating rules and an asynchronous update scheme at each discrete time step. Five external inputs representing ACSL4, ferroportin, p53, SCD1, and TFRC, which do not change during the course of the simulation, are used to study the sensitivity of ferroptosis induction to different signaling and perturbation conditions.

Konstorum et al. used a system-level analysis to study how different input conditions and parameters alter ferroptosis sensitivity. They found that ferroptosis sensitivity depends on PUFA synthesis and PUFA incorporation into the phospholipid membrane, as well as the balance between levels of pro-oxidant species (ROS, lipoxygenases) and antioxidant factors (GPX4). Ferroptosis sensitivity can be reduced by altering parameters that minimize the production of L-OOH species. High ACSL4 and low SCD1 levels result in high ferroptosis sensitivity. The model also predicted that a high level of SCD1 can inhibit ferroptotic induction even when levels of ACSL4 are high. These model predictions were validated using an in vitro experimental system of an ovarian cancer stem cell culture [10]. Overall, the model allows us to better understand the crosstalk between pathways transmitting signals from different inputs that induce the execution of ferroptosis.

## 6. Immunogenic Cell Death

Immunogenic cell death (ICD) is a regulated cell death mechanism that induces an immune response in the hosts [14]. Basically, ICD is an immunostimulatory form of apoptosis that is characterized by the ability of dying cells to generate robust adaptive immune responses [85]. The immune response is promoted by damage-associated molecular patterns (DAMPs), which are released by dying cells [86]. DAMPS communicate a state of danger to the organism by activating pattern recognition receptors (PRRs) that are present on the surface of innate immune cells such as monocytes, macrophages, and dendritic cells (DCs) [87]. Activated macrophages and dendritic cells can migrate to the lymph node and pass the antigens to CD8^+^ and CD4^+^ T lymphocytes, which results in an adaptive immune response. Tumor cell systems are often used to study ICD regulation and dynamics [88]. The immune responses against cancer- or pathogen-driven antigens that induce ICD are well characterized [85]. Importantly, over the past years, developments of ICD-related cancer immunotherapy approaches are gaining great momentum [89].

To study the ICD dynamics of cancer cells, Checcoli et al. developed a mathematical model that integrates intracellular mechanisms involved in ICD and intercellular interactions among cancer cells, DCs, CD8^+^, and CD4^+^ T cells [11]. The modeling approach is based on a continuous time Boolean Kinetic Monte-Carlo formalism that was successfully applied to model different complex molecular mechanisms [90]. The aim of the mathematical characterization of ICD processes was to identify the regulatory molecular targets and combinations of pharmacological compounds that can increase anticancer immunity. The model can predict the time-dependent size of different cell populations involved in ICD that is induced by a treatment exposure. 

To determine the role of each of the main cell types involved in ICD, Checcoli et al. first simulated a core ICD mechanism that is merely sufficient to reproduce ICD events observed experimentally [11]. The core regulatory mechanism describes the release of CALR, ATP, and HMGB1 molecules from dying tumor cells, and inner-state activation or evolution of immature DC, activated DC, migrating DC, lymph node DC, T cell, and cytotoxic T lymphocyte cell types. As shown in Figure 5, also included are two processes: tumor cell division, which is inhibited by T cells, and death, which is promoted by cytotoxic T lymphocytes. The states of molecules and cells are described by Boolean variables that can take two values: **1** for active or present and **0** for inactive or absent. The system state is described by a vector of Boolean values that represent each molecule, process, and cell type in the system. In the probabilistic description, the probability is assigned to each system state, such that the sum of probabilities over all possible system states is equal to **1**. Then, to determine the number of cells in a given system state, the system state probability is multiplied by the overall size of the cell population.

The core model can reproduce the series of events following an ICD-inducing intervention. The release of CALR, ATP, and HMGB1 molecules by dying cancer cells occurs within hours, a slow increase in T cells begins after 100 h, which peaks at 200 h, and the tumor cells are eliminated in about 220 h when a rapid increase in cytotoxic T lymphocyte cell population begins. When the clonal expansion of the cytotoxic T lymphocytes was blocked in the model, tumor cell clearance became less efficient and depended mostly on the direct cytotoxicity of the treatment.

To improve the predictive power of the model, Checcoli et al. extended their core model by including more cell types and molecular components as well as the ligand–receptor dynamics that determines intercellular communication. The extended model describes 57 entities and provides more detailed representations of the series of events that were explored by the core model. Simulation results of the extended model also reproduce the succession of events resulting in ICD. Simulations were performed starting with 80% of tumor cells, 10% of dendritic cells, and 5% of inactive CD4^+^ and CD8^+^ cells. The population of tumor cells rapidly decays starting from 250 h when cytotoxic T lymphocytes are engaged to eliminate tumor cells. 

To assess the extended model robustness to parameter changes, Checcoli et al. performed a sensitivity analysis measuring the variations in sizes of tumor cell populations within the 220 h and 280 h time frame when the tumor cell population decreases in the standard conditions (WT) of the extended model [11]. The decrease in size of the tumor cell population was seen to be delayed only for a few parameter changes when compared with the WT condition. Changes in parameters that control the number of DCs gave the strongest effect. A lower amount of DCs delayed the time of death, whereas a higher amount enhanced the death process. Changes in parameters controlling the rate of T cell clonal expansion give a similar effect on the cell death process. Sensitivity analysis also suggested the points of intervention that had the strongest effect on ICD. For example, a complete knockout of CD28 or CD80 (costimulatory molecules for T cell activation) resulted in a failure of the ICD-inducing treatment (80% of the tumor cell population persists at t = 280 h). By contrast, an external treatment that increases Interleukin-2 (IL-2) could kill the tumor cells faster, at t = 200 h.

The Boolean approach does not provide quantitative details and different regimens of drug treatments. Nevertheless, the model characterizes ICD events and dynamics in cancer cells and predicts molecular targets that could increase tumor clearance. For future directions, Checcoli et al. suggested to include specific in vitro and in vivo experiments to identify parameter values that will agree with experimentally observed timing of the different events leading to tumor clearance [11]. Further extension of the model including effects of IFNγ or TGFβ on the immune cells, and major signaling pathways inside each cell type, will allow the model to predict more feasible pharmacological interventions that can boost ICD for killing tumor cells.

## 7. Discussion

The significant progress that has been made in the mathematical characterization of different cell death execution pathways offers quantitative insight into cell death control and mechanistically explains why and how a living cell may die. Table 1 summarizes cell death mathematical model development over a 22-year period. We include the modeled cell death mechanism, methods, a mathematical description of the cell death event used in each model, and the main modeling results obtained in each work. ODE and Boolean logic-based approaches are the most common mathematical techniques used to model cell death mechanisms. However, a physical description based on the potential landscape theory has been recently applied to study stochastic dynamics and global stability of cell death signaling pathways [8,73]. In this approach, the steady state probability distribution of a system *P_ss_* and a dimensionless potential function *E* are related via Boltzmann relation: *E* = −ln(*P_ss_*) [91]. The physical description allows one to employ thermodynamics to analyze cell death regulatory circuits. Conversely, entropy-based approaches have been applied to analyze biological networks [92] and a cell fate selection process [93,94], they have not been yet applied to characterize cell death decision mechanisms. Therefore, one promising future direction is to describe cell death networks using physical approaches that could help to reveal new functional system states and unknown properties of cell death regulatory mechanisms.

Importantly, many different cell death pathways share common molecular components, and thus all these pathways can interact together at any time to form a complex mechanism. Therefore, we hypothesize that cell death can be controlled by a singular, highly integrated cell death decision network, see Figure 6. This network enables cells to alter the signal flow through the shared nodes but with different edges and so select alternative cell death execution pathways within a single control network of cell death. A stress death signal can thus initiate multiple death mechanisms but not all reach an execution threshold. Currently, the molecular mechanism that regulates the selection of each specific death execution pathway remains elusive. In addition, mathematical models developed to study crosstalk between necroptosis and apoptosis [65], pyroptosis and apoptosis [73], autophagy and apoptosis [29] support the hypothesis that signals propagating through different cell death pathways are integrated to process the execution of specific cell death. We are developing a mathematical model of the cell death decision network to predict the molecular species and interactions that direct the signal flow towards a specific irreversible cell death fate. Such a model will provide new insights into the integrated control of cell death. Model predictions will help develop new approaches to either block or initiate irreversible cell death and identify which cell death pathways are blocked and which pathways remain accessible to execute cell death. Thus, model predictions will suggest alternative interventions to overcome a block in cell death activation that can occur in cancer cells that acquire drug resistance.

## Figures and Tables

**Figure 1 entropy-24-01402-f001:**
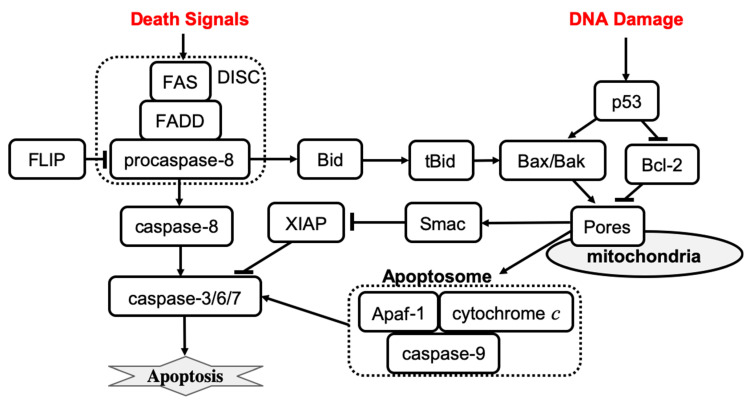
The mechanisms of receptor-induced apoptosis (left) and stress-mediated, mitochondria-dependent apoptosis (right). Solid lines represent activation (arrowhead) and inhibition (bar head) influences.

**Figure 2 entropy-24-01402-f002:**
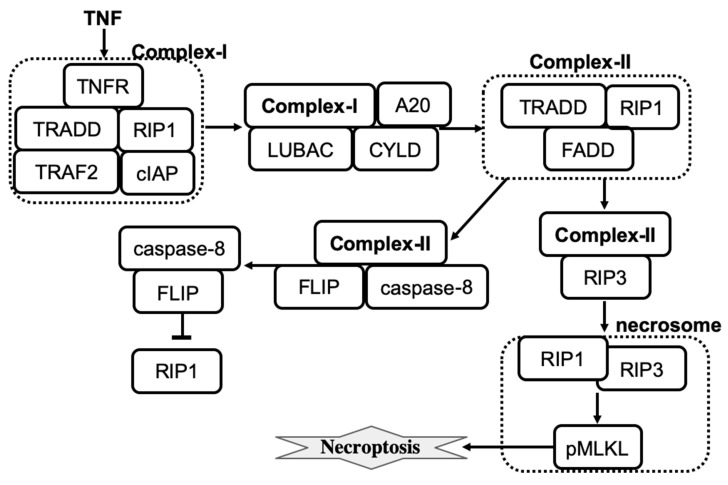
The influence diagram shows key molecular components involved in necroptosis regulation. Lines represent activation (arrowhead) and inhibition (bar head) influences.

**Figure 3 entropy-24-01402-f003:**
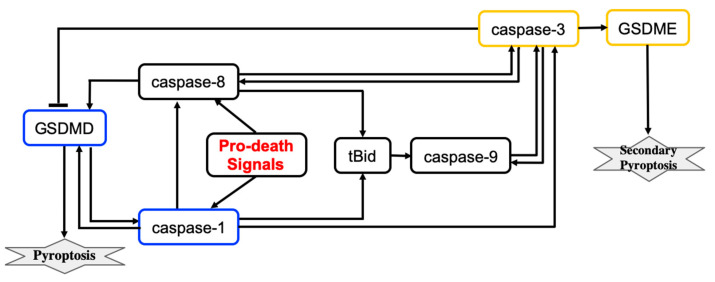
The influence diagram shows key molecular components involved in GSDMD- and GSDME-executed death modes called pyroptosis and secondary pyroptosis respectively. Lines represent activation (arrowhead) and inhibition (bar head) influences.

**Figure 4 entropy-24-01402-f004:**
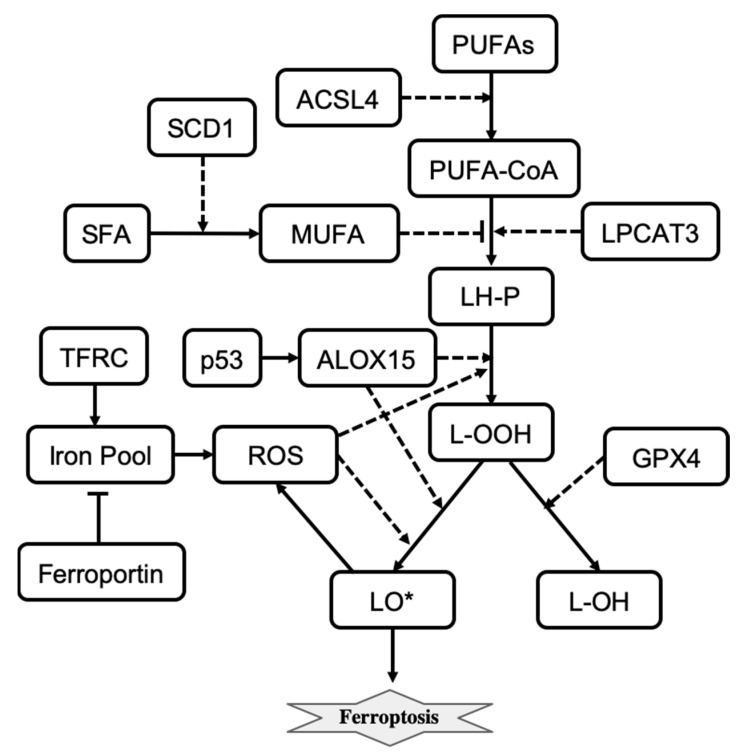
The influence diagram shows key molecular components involved in ferroptosis regulation. Solid arrows represent reactions of transformation, activation (arrowhead), and inhibition (bar head). Dashed lines with arrowheads or bar heads represent activation or inhibition of a reaction, respectively.

**Figure 5 entropy-24-01402-f005:**
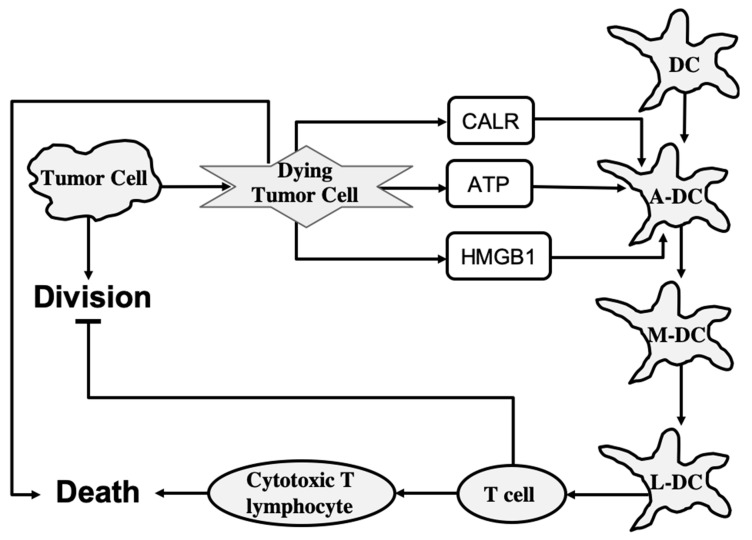
The core mechanism of immunogenic cell death. Lines represent activation (arrowhead) and inhibition (bar head) influences. A-DC, M-DC, and L-DC represent activated DC, migrating DC, and lymph node DC cell types, respectively.

**Figure 6 entropy-24-01402-f006:**
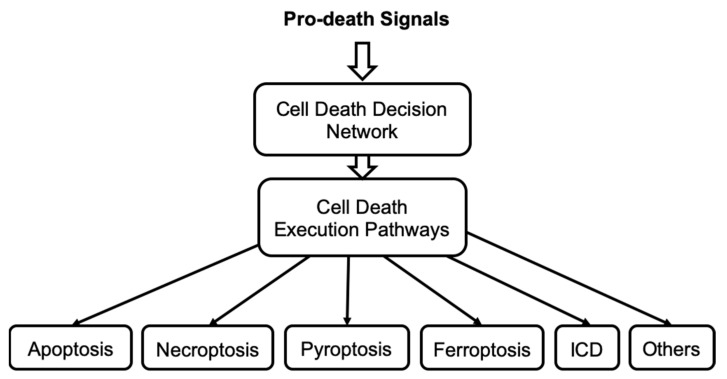
The concept of cell death decision network that controls the irreversible execution of different cell death mechanisms.

**Table 1 entropy-24-01402-t001:** Summary of cell death mechanism models.

Authors, Year, Cell Death Mechanism Modeled	Methods. Death Rule (DR)	Results
Fussenegger et al., 2000 [7], receptor- and stress-induced apoptosis	ODE approach. DR: the ratio of executioner caspase to free Bcl-x_L_ is greater than a threshold value	Qualitative explanation of observed caspase activation dynamics
Eissing et al., 2004 [21], receptor-induced apoptosis	ODE approach, stability and bifurcation analysis methods. DR: The bistable system is in apoptotic steady state	Bistable behavior of caspase-3 activation
Bentele et al., 2004 [25], receptor-induced apoptosis	ODE approach, sensitivity analysis. DR: receptor–ligand ratio is greater than a threshold value	A threshold mechanism for induction of receptor-induced apoptosis
Hua et al., 2005 [43], receptor-induced apoptosis	ODE approach, sensitivity analysis. DR: caspase-3 activation	Bcl-2 blocks the mitochondrial apoptosis pathway by binding to proapoptotic proteins
Legewie et al., 2006 [22], intrinsic apoptosis	ODE approach, stability and bifurcation analysis methods. DR: irreversible caspase-3 activation	Bistable and irreversible caspase-3 activation arises in the system due to XIAP-mediated feedback
Rehm et al., 2006 [26], intrinsic apoptosis	ODE approach, sensitivity analysis. DR: complete caspase-dependent substrate cleavage	All-or-none apoptotic response depends on caspase-3-dependent feedback signaling and XIAP
Bagci et al., 2006 [23], mitochondria-dependent apoptosis	ODE approach. DR: caspase-3 activation is above a threshold that depends on Bax degradation and expression rates.	The transition from bistable to monostable (survival) cell behavior is controlled by the number of mitochondrial permeability transition pores
Chen and Cui et al., 2007, 2008 [38,45,46], intrinsic apoptosis	Deterministic and stochastic approaches, robustness analysis. DR: one-way bistable switch of Bax-activation	Apoptotic switches are bistable and robust to noise
Albeck et al., 2008 [27], extrinsic apoptosis	ODE approach, compartmental modeling. DR: mitochondria-to-cytosol cytochrome *c* and Smac translocation in an all-or-none manner	Permeabilization of the mitochondrial membrane and relocalization of proteins are the key factors in all-or-none death decision
Spencer et al., 2009 [33], extrinsic apoptosis	ODE approach. DR: levels of activated tBid, Bax, and Bak exceed a threshold set by inhibitory Bcl-2 proteins	Cell-to-cell variability in time-to-death depends on activation of the pore-forming proteins Bax and Bak
Neumann et al., 2010 [32], crosstalk between receptor-mediated apoptosis and NF-κB signaling	ODE approach, sensitivity analysis. DR: the maximum level of active caspase-8 is used as a readout for apoptosis	Assembly of DISC acts as a signal processor determining life/death decisions in a nonlinear manner
Hong et al., 2012 [28], crosstalk between apoptosis and ER stress response mechanisms	ODE approach, sensitivity analysis. DR: the level of apoptosis is determined by an ODE that depends on caspases-2,3,9,8 and apoptosis-inducing factor	Crosstalks among the mitochondrial, death receptor and ER stress response pathways contribute to the level of apoptosis
Tavassoly et al., 2015 [29], crosstalk between autophagy and apoptosis	ODE approach. DR: apoptosis occurs as soon as proapoptotic BH3 exceeds antiapoptotic Bcl2 protein	Time courses of the relative level of autophagy for different levels of stressor and percentage of apoptotic cells
Ballweg et al., 2017 [30], crosstalk between p53 signaling and apoptosis	ODE approach, dynamical analysis. DR: the level of p53 is elevated higher than a threshold that depends on cIAP level	The probability of apoptosis depends on the dynamics of p53
Schlatter et al., 2009 [34], apoptosis	Boolean logic and multi-value logic approach	High connectivity, crosstalks, and feedback loops in apoptosis regulatory network are significant and essential for apoptosis signaling
Mai et al., 2009 [35], intrinsic and extrinsic apoptosis	Boolean logic approach. DR: the “DNA Damage Event’’ node has remained in the ON state for 20 successive steps	The feedback loops directly involving the caspase 3 are essential for maintaining irreversibility of apoptosis
Calzone et al., 2010 [36], apoptosis and non-apoptotic cell death (necroptosis)	Boolean logic approach. DR: “Apoptosis” node or “NonACD” node is in ON state	Transient activation of key proteins in necroptosis and mutual inhibitory crosstalks among apoptosis, survival and necroptosis pathways
Xu et al., 2021 [8], cellular necroptosis signaling circuits	ODE approach, sensitivity analysis, bifurcation and potential landscape methods.	The structure and distribution characteristics of all parameters are essential for stable oscillation behavior of necroptosis circuits
Ildefonso et al., 2022 [59], necroptosis regulation	ODE approach, DREAM parameter estimation method, sensitivity analysis. DR: phosphorylated MLKL exceeds a hard threshold of 2772 molecules	Four distinct necroptosis execution modes
Li et al., 2021 [65], crosstalk between apoptosis and necroptosis regulatory networks.	ODE approach. DR: apoptosis occurs when RIP1 level < ∼1000 molecules/cell, co-occurrence of apoptosis and necroptosis when ∼46,000 mpc< RIP1 > ∼1000 mpc, necroptosis alone when RIP1 >∼46,000 mpc	Characterization of RIP1’s biphasic roles in necroptosis
Zhu et al. [9], crosstalk between caspase-1 and caspase-3 driven pyroptosis pathways	ODE approach, bifurcation and sensitivity analysis methods. DR: Cell death rate is defined using a ratio of dying cell population to the initial cell population	The change in expression levels of caspase-1, caspase-3, and GSDMD can switch between GSDMD- and GSDME-executed pyroptosis death modes
Li et al., 2022 [73], crosstalk between pyroptosis and apoptosis regulations	ODE and potential energy landscape approaches. DR: by levels of cleaved GSDMD (pyroptosis) and cleaved caspase-3 (apoptosis)	Caspase-1 and GSDMD are key proteins that regulate the switching between pyroptosis and apoptosis
Konstorum et al., 2020 [10], ferroptosis regulation	Stochastic, multistate, discrete mathematical approach. DR: intermediate and high levels of the lipid radical LO*	Ferroptosis sensitivity depends on PUFA synthesis, PUFA incorporation into the phospholipid membrane, and the balance between levels of pro-oxidant species and antioxidant factors
Checcoli et al., 2020 [11], immunogenic cell death (ICD) mechanism	Boolean Kinetic Monte-Carlo approach. DR: Death node is at **1**	The succession of events resulting in ICD. Points of intervention that had the strongest effect on ICD

## Data Availability

Not applicable.

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
