# Peer review of "Mathematical Models of Death Signaling Networks"

_entropy, 2022, doi:10.3390/e24101402_

Round 1

Reviewer 1 Report

The manuscript titled "Mathematical Models of Death Signaling Networks" presents a comprehensive review of the cutting-edge research progress in cell death signaling made by mathematical modeling. Over the past decade, computational analysis provides many meaningful insights into the regulatory mechanisms underlying the complicated death fate decision of cells. The manuscript is informative and well-structured. The present manuscript may deliver results that will surely be of interest to the readership and I warmly recommend it for publication in Entropy. There are still some issues with the manuscript that merit discussion.

1)    The authors review many mathematical models for each mode of cell death, but the logical relationship between these models is not clear. Could you summarize the development process of the mathematical model of cell death in the form of a table, and describe the corresponding methods and main conclusions?

2)    Recent studies have shown that reduction of caspase-1 or GSDMD switches cell death from pyroptosis to apoptosis, and only the expression levels of caspase-1 and GSDMD have the potential to individually switch cell death modes. It is recommended that the authors refer to Research, 2022, 9838341, 17(2022) to discuss the mathematical model of pyroptosis in this manuscript.

3)    The potential energy landscape explains the global dynamics of mode switching in crosstalk regulatory pathways between different cell death modes. Could the authors provide some forward-looking guidelines on the mode switching of cell death from the perspective of physics such as energy and entropy in the Discussion section to closely follow the theme of this Journal.

Author Response

The manuscript titled "Mathematical Models of Death Signaling Networks" presents a comprehensive review of the cutting-edge research progress in cell death signaling made by mathematical modeling. Over the past decade, computational analysis provides many meaningful insights into the regulatory mechanisms underlying the complicated death fate decision of cells. The manuscript is informative and well-structured. The present manuscript may deliver results that will surely be of interest to the readership and I warmly recommend it for publication in Entropy. There are still some issues with the manuscript that merit discussion.

We thank the Reviewer for endorsing our work and making helpful suggestions for improving the manuscript.

1)    The authors review many mathematical models for each mode of cell death, but the logical relationship between these models is not clear. Could you summarize the development process of the mathematical model of cell death in the form of a table, and describe the corresponding methods and main conclusions?

This is a great suggestion. We have added a table (Table 1) that provides the summary of cell death models, methods and model results.

2)    Recent studies have shown that reduction of caspase-1 or GSDMD switches cell death from pyroptosis to apoptosis, and only the expression levels of caspase-1 and GSDMD have the potential to individually switch cell death modes. It is recommended that the authors refer to Research, 2022, 9838341, 17(2022) to discuss the mathematical model of pyroptosis in this manuscript.

Thank you for bringing up the relevant work. We have added discussion of this model to the pyroptosis section of the review.

3)    The potential energy landscape explains the global dynamics of mode switching in crosstalk regulatory pathways between different cell death modes. Could the authors provide some forward-looking guidelines on the mode switching of cell death from the perspective of physics such as energy and entropy in the Discussion section to closely follow the theme of this Journal.

We have added the physical approaches and the potential landscape theory discussion to Discussion section.

Reviewer 2 Report

In the introduction, I expected to find a mathematical definition of what "cell death" is and how "death" is described in terms of a theoretical model. What is the proper way to identify a state of irreversible fate of cell death? The review discusses in detail the various forms of cell death, the regulation, and the critical molecular player. The description of processes is worthwhile to read for biologists. He will find all the relevant terms, molecules and complexes that are broadly discussed in the experimental literature. For a theoretician the value of the review may be not very relevant. I doubt that, the incorporation of more and more details in the models is the best way to proceed as long as the theoretical framework and question to answer is not well defined. Modeller may apply various techniques to find "interesting" results. Whether the theoretical modeling contributes to understand "cell death" remains an open question for me. My personal preferences to see an discussion of the terminology and theoretical framework and not a discussion of the applied simulation techniques and biological details, does, however, not lower the value of the review of the authors.

Author Response

In the introduction, I expected to find a mathematical definition of what "cell death" is and how "death" is described in terms of a theoretical model. What is the proper way to identify a state of irreversible fate of cell death?

We thank the Reviewer for this suggestion. We have added the following paragraph to the introduction section:

“Cell death execution is an all-or-none, irreversible process [20]. Mathematically the activation of irreversible cell death can be described by an irreversible bistable switch with a stable survival steady state, a stable death steady state, and a third unstable steady state separating the survival and death states [21-24]. A pro-death signal can induce cell death by driving the bistable system from the survival to the death state. The transition occurs when the pro-death signal reaches a threshold that corresponds to the limit point bifurcation. Transition in the reverse direction, from death to survival, is impossible because the second limit point bifurcation, where the death steady state vanishes, occurs in the biologically irrelevant negative signal values (i.e. the concentration of a death-inducing ligand or stressor cannot be negative). Therefore, the activation of the cell death execution in such bistable system cannot be reversed even if the initial cell death trigger is removed. This mathematical description of the cell death activation is consistent with a threshold mechanism for cell death induction [25] and an all-or-none death decision [22, 26, 27]. Importantly, understanding how cells control the cell death/survival switch can help to identify targets that can force cancer cells to flip the switch to activate the irreversible cell death execution.”

The review discusses in detail the various forms of cell death, the regulation, and the critical molecular player. The description of processes is worthwhile to read for biologists. He will find all the relevant terms, molecules and complexes that are broadly discussed in the experimental literature. For a theoretician the value of the review may be not very relevant. I doubt that, the incorporation of more and more details in the models is the best way to proceed as long as the theoretical framework and question to answer is not well defined. Modeler may apply various techniques to find "interesting" results. Whether the theoretical modeling contributes to understand "cell death" remains an open question for me. My personal preferences to see an discussion of the terminology and theoretical framework and not a discussion of the applied simulation techniques and biological details, does, however, not lower the value of the review of the authors.

We intended to write a review not only for modelers but also for biologists. We agree with Reviewer that a methodological review could be indeed more relevant for theoreticians but would be less attractive for experimentalists and biologists.

We have added a table in discussion which summarizes cell death models, methods and model results, and also provide a specific “cell death rule” that is used in each model to determine the cell death execution event. The table will help readers to find relevant information and sources where they can find more details.

We thank the Reviewer for the constructive comments.